# Chronological Changes in the Expression and Localization of Sox9 between Achilles Tendon Injury and Functional Recovery in Mice

**DOI:** 10.3390/ijms241411305

**Published:** 2023-07-11

**Authors:** Genji Watanabe, Masahito Yamamoto, Shuichirou Taniguchi, Yuki Sugiyama, Hidetomo Hirouchi, Satoshi Ishizuka, Kei Kitamura, Toshihide Mizoguchi, Takashi Takayama, Katsuhiko Hayashi, Shinichi Abe

**Affiliations:** 1Department of Anatomy, Tokyo Dental College, 2-9-18 Kandamisaki-cho, Chiyoda-ku, Tokyo 101-0061, Japan; watanabegenji@tdc.ac.jp (G.W.); yamamotomasahito@tdc.ac.jp (M.Y.); taniguchisyuuichirou@tdc.ac.jp (S.T.); sugiyamayuuki@tdc.ac.jp (Y.S.); hirouchihidetomo@tdc.ac.jp (H.H.); 2Department of Pharmacology, Tokyo Dental College, 2-9-18 Kandamisaki-cho, Chiyoda-ku, Tokyo 101-0061, Japan; ishidukasatoshi@tdc.ac.jp; 3Department of Histology and Developmental Biology, Tokyo Dental College, 2-9-18 Kandamisaki-cho, Chiyoda-ku, Tokyo 101-0061, Japan; kitamurakei@tdc.ac.jp; 4Oral Health Science Center, Tokyo Dental College, 2-9-18 Kandamisaki-cho, Chiyoda-ku, Tokyo 101-0061, Japan; tmizoguchi@tdc.ac.jp; 5Department of Dentistry, The Jikei University School of Medicine, 3-19-18 Nishi-shinnbashi, Minato, Tokyo 105-8471, Japan; tt1110@jikei.ac.jp (T.T.); katsuh@jikei.ac.jp (K.H.)

**Keywords:** Sox9, tendon, regenerative medicine, myotendinous junction, skeletal muscle, cartilage

## Abstract

Tendons help transmit forces from the skeletal muscles and bones. However, tendons have inferior regenerative ability compared to muscles. Despite studies on the regeneration of muscles and bone tissue, only a few have focused on tendinous tissue regeneration, especially tendon regeneration. Sex-determining region Y-box transcription factor 9 (Sox9) is an SRY-related transcription factor with a DNA-binding domain and is an important control factor for cartilage formation. Sox9 is critical to the early-to-middle stages of tendon development. However, how Sox9 participates in the healing process after tendon injury is unclear. We hypothesized that Sox9 is expressed in damaged tendons and is crucially involved in restoring tendon functions. We constructed a mouse model of an Achilles tendon injury by performing a 0.3 mm wide partial excision in the Achilles tendon of mice, and chronologically evaluated the function restoration and localization of the Sox9 expressed in the damaged sites. The results reveal that Sox9 was expressed simultaneously with the formation of the pre-structure of the epitenon, an essential part of the tendinous tissue, indicating that its expression is linked to the functional restoration of tendons. Lineage tracing for Sox9 expressed during tendon restoration revealed the tendon restoration involvement of cells that switched into Sox9-expressing cells after tendon injury. The stem cells involved in tendon regeneration may begin to express Sox9 after injury.

## 1. Introduction

A tendon is an important piece of dense connective tissue for transmitting forces from the skeletal muscles to the bones. Accordingly, tendinous tissue is tissue in the musculoskeletal system and is the linchpin of physical functions for moving and stabilizing the body. In recent years, more patients have presented with tendon injuries, with 110 million patients in the USA alone [1,2,3]. As such, concerns have been raised, as decreased physical functions in patients with a tendon injury may lead to a decline in their quality of life [4].

A tendon consists of groups of collagen-rich fascicles with an interfascicular matrix that fills the gaps between the fascicles. It is externally covered with epitenon and paratenon. The fascicles consist of fibers, each of which comprises a combination of collagen type 1 alpha 1 chain (Col1a1) and collagen type 1 alpha 2 chain at a ratio of 2:1 [5]. The regenerative ability of mature tendons is inferior to that of muscles and bones. This is attributed to the low ratio of cells to matrices and the scarcity of blood vessels, which makes the distribution of oxygen and nutrients to damaged sites difficult [6,7,8]. Although reconstruction surgery may be performed for treating damaged tendons, the complication risk is high, and postoperative recurrences may increase because of the poor self-repair ability. Thus, no appropriate therapeutic approaches have been established [9]. Accordingly, we believe that revealing the regenerative mechanism of tendons holds the key to repairing and maintaining motor functions and treating and preventing new injuries.

Various associations between tendon development and regeneration have been reported; accordingly, understanding tendon development is considered an essential factor in identifying more efficient and scientific treatments for tendon injuries [10]. Studies have reported that scleraxis (Scx) is expressed in the tendon progenitor cells at the earliest stage of development [11], followed by the expression of tenomodulin (Tnmd) and Mohawk (Mkx), which promote tendon maturation [12,13]. The sex-determining region Y-box transcription factor 9 (Sox9) protein is concurrently expressed alongside Scx from the early-to-middle stage of tendon development [14]. Tendon cells are formed when Sox9-expressing mesenchymal cells differentiate into tendon progenitor cells [15]. However, the expression of Sox9 decreases in the late stage of development [14].

Scx is an important marker of tendon regeneration, as is the case of their development [11,16]. In addition, alpha-smooth muscle actin (α-SMA) is involved in activating these Scx-positive cells to accumulate defects [17]. Tnmd is regulated in tendon regeneration by suppressing heterotopic ossification during tendon injury. Moreover, Tnmd maintains a balance between the tendon lineage and myofibroblastic cells [18]. The extracellular matrix (ECM)-associated protein periostin (Postn) was also identified to contribute to the maintenance of tendon stem/progenitor cell function and promote tendon regeneration [19]. The hedgehog signaling pathway in tendon development also plays a crucial role in their regeneration [20]. Thus, the molecular regulators identified in studies of tendon development were also reported to be involved in repair mechanisms in tendon injury [21].

Sox9-positive osteochondral progenitor cells differentiate into chondrocytes, osteoblasts, and osteocytes during fracture repair [22]. In addition, the hedgehog coreceptor Smoothened is significantly involved in the differentiation of Sox9 into cartilage and bone during bone regeneration [23]. Although the regenerative processes of ligaments and enthesis have been assessed in the past, few studies have examined the involvement of Sox9 in tendon regeneration [24,25,26]. Much remains unknown about the timing of Sox9 expression and its localization during healing in tendons. In this study, we constructed a tendon injury mouse model and chronologically evaluated the localization of Sox9 to uncover changes in Sox9 expression from the onset of tendon injury until functional recovery. Moreover, we studied Sox9-positive cells that accumulated during tendon restoration to reveal temporal relationships.

## 2. Results

### 2.1. Functional Torque Test

The physiological test revealed a significant decline in functions at 1 week after the operation (POW1) compared with that observed in the sham. At 2 weeks after operation (POW2), the animals exhibited a significant increase in function compared with 1 week after the operation. In turn, at 4 weeks after the operation (POW4), a significant increase in function was observed compared with 2 weeks after the operation. No significant functional differences were noted between the sham and the 4 weeks after the operation settings (Figure 1C).

### 2.2. Morphological Analysis

First, in the epitenon region, the continuity of the epitenon was ruptured at 1 week after the operation (Figure 2A,B), showing an increase in cell density compared with the sham (Figure 2E,F,M). At 2 weeks after the operation, the connective and tendon tissue boundaries were more distinct than those detected 1 week after the operation; however, no formation of the pre-structure of the epitenon was observed (Figure 2C). At 4 weeks after the operation, the injury site was covered with the pre-structure of the epitenon, as in the sham, and tendon sheath formation was detected (Figure 2A,D; arrows).

In the inner part of the injury site, a significant increase in cell count was observed 1 week after the operation vs. the sham. Compared with 1 week after the operation, the cell density in the injury site was significantly decreased at 2 weeks and at 4 weeks after the operation (Figure 2B–D,F–H,M). The cell density differed significantly between the sham and 4 weeks after the operation, and the actual hematoxylin and eosin (H&E) staining images showed that the cells remained irregular regarding their morphology and arrangement (cell polarity) (Figure 2E,H). In addition, the tendon swelling caused by the injury, as assessed based on width diameter, caused an increase in the width of the tendinous tissue at all time points after the operation compared with the sham; moreover, the width change had not recovered at 4 weeks after the operation (Figure 2N).

Subsequently, Azan staining was performed to confirm the presence of collagen fibers at the site of injury. No collagen fibers were detected at the injury site at 1 week after the operation; in contrast, these fibers appeared at 4 weeks after the operation (Figure 2J–L). Although the sham was strongly stained with azocarmine red, the injury sites at 2 weeks and 4 weeks after the operation were strongly stained with aniline blue (Figure 2I,K,L). The collagen orientation disappeared at 1 week after the operation, with an irregular collagen orientation observed from 2 weeks after the operation and onwards. At 4 weeks after the operation, the collagen orientation became parallel to the longitudinal axis, similar to that observed in the sham (Figure 2I–L). The Achilles tendons were stained in blue in the sham, which was overstained (24 h) with aniline blue (Figure 2O).

The continuity of the epitenon, which is critical for tendon regeneration, had already disappeared at 1 week after the operation (Figure 2B). The cell density at the injury site peaked in this period, before decreasing in the subsequent period (Figure 2B,F,M). The pre-structure of the epitenon region achieved histological recovery at 4 weeks after the operation. However, the cell polarity and width of the tendinous tissue did not exhibit a similar trend to that of the sham at 4 weeks after the operation (Figure 2E,H).

### 2.3. mRNA Expression Triggered by Tendon Injury

The reverse-transcription polymerase chain reaction (RT-PCR)-based mRNA search revealed that Sox9 was significantly upregulated at 1 week after the operation and 2 weeks after the operation compared with the control (Figure 3A). However, no significant differences were noted among any of the remaining groups. α-SMA showed significantly higher values in the control, 1 week after the operation, and 4 weeks after the operation, with the values recorded at 4 weeks after the operation being higher than those noted at 1 week after the operation (Figure 3B). Regarding Postn, significant differences were more pronounced between 1 week after the operation and any of the remaining groups, with the difference with the control being the most significant (Figure 3C). In Col I, significant increases were noted at 1 week after the operation and 2 weeks after the operation compared with the control (Figure 3E). In terms of fibromodulin, which is a proteoglycan involved in collagen formation, significant increases were noted at 2 weeks after the operation compared with the control and 1 week after the operation (Figure 3G). Finally, regarding Scx and Col III, increasing tendencies were noted as the weeks elapsed after injury; however, no significant differences were observed among any of the groups.

### 2.4. Immunohistochemical Staining of Tendon Injury Samples

Immunohistological staining revealed that the expression of Sox9 in the cells started at the injury site 1 week after the operation (Figure 4B,F; red arrows). Similarly, α-SMA and Postn were expressed more strongly outside vs. within the injury site (Figure 4K,Y). At 2 weeks after the operation, Sox9-positive cells were primarily expressed at the tendon injury site (Figure 4C,G; red arrows). The expression levels of α-SMA and Postn were also more pronounced within the injury site (Figure 4L,V). Moreover, at 1 week after the operation, α-SMA, which is a marker of vascular endothelial cells, was expressed in the connective tissue outside the injury site. In contrast, at 2 weeks after the operation, it accumulated within the injury site (Figure 4K,L,P,Q), whereas at 4 weeks after the operation, in which functional recovery was achieved, Sox9-positive cells accumulated continuously within the injury site. Conversely, α-SMA was mainly expressed in the pre-structure of the epitenon region, exhibiting a similar tendency to the sham (Figure 4J,M,O,R). However, because a slight expression of this molecule was observed within the injury site, a complete recovery of polarity was not achieved.

Graphs showing the proportions of Sox9, α-SMA, and Postn at the injury site were created. Regarding Sox9, significant differences were noted at 2 weeks after the operation and 4 weeks after the operation compared with all other groups (Figure 4I). Significant differences in α-SMA were noted at 2 weeks after the operation compared with all other groups (Figure 4S). Significant differences in Postn were noted at 2 weeks after the operation compared with all other groups and at 4 weeks after the operation compared with the sham (Figure 4X). The expression of Sox9, α-SMA, and Postn was upregulated at 2 weeks after the operation compared with the sham. Thereafter, these proteins were significantly decreased at 4 weeks after the operation compared with 2 weeks after the operation. 

### 2.5. Lineage Tracing of Sox9-Expressing Cells

To trace the lineage of *Sox-9*-expressing cells during tendon injury, *Sox9-Cre-ERT2 and LSL-tdTomato* mice were hybridized to create *Sox9-Cre ERT2* and *tdTomato* mice (Figure 5A). The comparison of the group that was administered tamoxifen before the injury (“Pre” regimen) with the group that received the administration of tamoxifen after the injury (“Post” regimen) revealed that the number of *Sox9-Cre ERT2; tdTomato*^+^ cells was significantly increased in the Post group (Figure 5B,C). The proportion of *Sox9-Cre ERT2; tdTomato*^+^ cells at the injury site was significantly increased in the Post group compared with the Pre group (Figure 5D). Moreover, the area of the *Sox9-Cre ERT2; tdTomato*^+^ cells in the actual injury region was significantly increased in the Post vs. the Pre group (Figure 5E). These results revealed that the proportion of *Sox9-Cre ERT2*; *tdTomato*^+^ cells that accumulated at the injury site was significantly increased after injury compared with those observed before injury.

## 3. Discussion

We investigated whether Sox9 is an appropriate molecular target for functional recovery during tendon injury in adults. The results revealed that Sox9, which was not expressed in the normal tendinous tissue of adult animals, accumulated in the pre-structure of the epitenon region and injury site during tendon regeneration. The physiological functional test and histological search revealed that the timing of the functional recovery of tendon injuries coincided with the timing of the regeneration of the pre-structure of the epitenon. Previous studies have also reported that the epitenon is one of the potential sources of tendon stem/progenitor cells, which is an essential topic in research on the regeneration of adult tendons [27,28]. When the pre-structure of the epitenon region, which had been ruptured by tendon injury, was regenerated, the expression of Sox9 started within the pre-structure of the epitenon region, indicating that Sox9 expression increased over time in the injured area. The results of cell-lineage tracing indicated that the Sox9-expressing cells that accumulated at the injury site were not derived from cells that were originally present there; rather, the cells that accumulated at the injury site were Sox9-expressing cells.

The results of the physiological functional test revealed a significant decline in functions at 1 week after the operation, whereas functional recovery started at 4 weeks after the operation (Figure 1C). Another study that examined functional recovery after an Achilles tendon injury using a treadmill also reported that functional recovery was achieved after 4 weeks. Therefore, functional recovery is considered to be achieved after approximately 4 weeks [29]. A histological test was also performed to examine the density of cells that accumulated at the injury site; its results indicated the highest cell density and a significant functional decline at 1 week after the operation (Figure 1C and Figure 2M). Moreover, the pre-structure of the epitenon was restored at 4 weeks after the operation. Therefore, the regeneration of the pre-structure of the epitenon may be greatly involved in functional recovery (Figure 1C and Figure 2A,D). This is consistent with previous reports arguing that the epitenon plays an important role in tendon regeneration [27,28].

The results of immunohistological staining revealed that proportions of Sox9, α-SMA, and Postn at the injury site exhibited the strongest levels at 2 weeks after the operation (Figure 4I,S,X). By the time point at which the boundary between the tendon tissue and the connective tissue is well defined, there may be an influx of regeneration-related cells and factors into the injured area. As α-SMA is also a marker of myofibroblasts in tendons and ligaments, the formation of myofibroblasts observed at 2 weeks after the operation may have activated tendon regeneration [30]. The results of RT-PCR revealed that Postn related to blood vessel induction was most significantly increased at 1 week after the operation (Figure 3C) [31]. Although this finding contradicted the results depicted in Figure 4X, this figure merely reported the density within the injury site. As demonstrated by the immunohistological staining experiment, it was strongly expressed at 1 week after the operation (Figure 4Y). This suggests that blood vessels are involved in the activation and promotion of the stem cells that are required for the regeneration of muscles, bones, and tendons.

Despite the various studies on tendon development available in the literature, much remains unknown about the factors involved in the regeneration and maintenance of injured tendons in adults. Adult tendinous tissue has a poor regenerative ability because of the small number of cells and scarcity of blood vessels. Because of the scarcity of tendon-forming cells that accumulate during tendon injury, histologically, fibrous healing is achieved in adult tendons. Previous studies have reported that permanent scarring is mediated by α-SMA-expressing cells in in vivo experiments [29,32]. The present study also showed an absence of the regeneration of tissue or cell polarity; H&E staining revealed that the cell polarity remained lost 4 weeks after the operation compared with the sham (Figure 2E,H). Moreover, the results of Azan staining indicated that the sham group was strongly stained with azocarmine red, whereas at 4 weeks after the operation, the injured area was strongly stained with aniline blue (Figure 2I,L). It is commonly assumed that Achilles tendons, which are normally composed of collagen fibers, are stained blue; however, Lee et al. reported that the original Achilles tendon stained red [33]. Therefore, we suspect that the blue staining observed at 4 weeks after the operation may represent collagen fibers that have formed scar tissue. As a result, the original dense connective tissue was not restored. The absence of histological regeneration poses a problem; however, fibrous scarring can be reduced when transforming growth factor (TGF)-β signaling inhibition is mediated by Smad3 deletion. However, at the cost of this process, the functional regeneration of tendons will not proceed. Therefore, scarring is not necessarily a detrimental phenomenon [34,35].

Recent studies have reported that α-SMA-expressing cells activate Scx-positive cells and repair deletion during patellar tendon injury. Furthermore, during Achilles tendon injury, stem cell antigen-1(Sca-1)-positive and Scx-negative cell populations differentiate into Scx-positive tendinous cells, migrate into the injury site, and produce an ECM, thus playing an important role in tendon repair [16,17,28]. Scx- and Sox9-positive cells form tendons during tendon development, and these Sca-1-positive and Scx-negative cell populations differentiate into Scx-positive and Sox9-positive cells, respectively. When expressed at the injury site, they may maintain the functions and structure of the injury site [36,37]. Because the stem cells that accumulated in the injured area in this study switched to a Sox9-positive presentation after injury, the investigation of the relationship between the Sca-1-positive and Scx-negative cell populations and Sox9 may be an important future task.

According to Ackerman (2022), the cells that accumulate after tendon injury are classified into the fibroblastic cell group, ECM group, and tendinous fibroblast group. Only cells in the ECM group are positive for Scx, with the factors involved in the cells making up the other groups remain unknown [38]. In this study, Sox9, which accumulated at the injury site, may have been associated with any of the three groups. Because Scx is co-expressed during tendon development, accompanied by Smad3 (which is related to Sox9) expression, it may likely be associated with the ECM group. Sox10, which is part of the same family as Sox9, is expressed in the tendinous fibroblast group; thus, it may have some effect. Although cell groups that modify tendon injuries have gradually been identified, their origins have not been uncovered. However, we hypothesized that two cell types function as important pathways, i.e., blood vessel- and epithelial-derived cells. This issue needs to be swiftly resolved. In recent years, research reports have been published from various perspectives on the attachment of muscles to bones [39,40,41,42]. We also believe that we need to consider tendon injuries from a new perspective other than Sox9.

The occurrence of heterotopic ossification and ectopic cartilage formation are other problems associated with tendon regeneration in adults. These phenomena have also been reported in studies focusing on tendon injuries [16,29]. Sox9 is essential for cartilage formation. When Sox9 is excessively expressed, ectopic cartilage is generated in tendinous tissue; thus, Sox9 is highly likely to be involved in ectopic cartilage formation [43]. However, it is unlikely that Sox9 is the sole cause of ectopic cartilage formation caused by tendon injury. Although Mohawk suppresses cartilage differentiation, its expression may be downregulated during injury. This downregulation might result in cartilage formation in tendons [44]. Another possibility is the effect of mechanical stress. Some studies have reported that fibrous cartilage may be induced when compressive force is applied to tendons [45,46]. These reports have indicated that the formation of ectopic cartilage in adult tendon injuries is highly likely to be modulated by Sox9 and other factors. Moreover, after differentiation into ectopic cartilage, Scx expression is strongly reactivated. This suggests that ectopic cartilage may also be a part of the healing process [29]. The search for factors and chronological proteins involved in ectopic cartilage formation will be a future challenge.

The most interesting result of the present study was that the number of tdTomato^+^ cells in the injured area was significantly increased in the Post compared with the Pre group (Figure 5D). This suggests, for the first time, that the accumulation of post-injury cells, rather than the accumulation of existing Sox9-positive cells in the injured area, may contribute to the functional regeneration of tendons by switching to Sox9-positive cells. Regeneration after rib and femur fractures has been reported to originate from a lineage in which the cells that accumulate after injury switch to a Sox9-positive status, rather than from existing Sox9-positive cells [23,47]. In the future, the clarification of the origin of these expression-switching Sox9-positive cells will play an important role in tendon regeneration.

In recent years, the incidence of tendon injuries has been increasing [1,2]. In this study, 6-week-old mice were used as experimental animals to investigate the histological response to injury. Future studies should use mature mice to maintain motor functions in an aging society. The present results indicate that Sox9 may play an important role in functional recovery during tendon injury, and that cells that switched to a Sox9-expressing status accumulate after a tendon injury. The identification of the origin and signaling pathway of Sox9 will help understand the process of tendon regeneration.

## 4. Materials and Methods

### 4.1. Experimental Animals

Five 6-week-old male C57BL/6J mice (Sankyo Labo Service Corporation, Inc., Tokyo, Japan) were assigned to each group. In the present experimental group, both legs were operated upon or sham surgery was performed. Initially, functional tests were performed on the left leg of the mice in deep anesthesia. The mice were then euthanized with CO2 poisoning and the left leg was harvested for PCR and the right leg was harvested for histological analysis. All mouse experiments were conducted in adherence to the EU Directive 2010/63/EU, the National Research Council’s Guide for the Care and Use of Laboratory Animals, and the ARRIVE guidelines. The study was approved by the Institutional Animal Care and Use Committee of Tokyo Dental College (Protocol no. 220104). Sox9creER (STOCK Tg[*Sox9cre/ERT2*]1Msan/J) mice and *R26tdTomato* (B6;129S6-Gt[ROSA]26Sortm14[CAG-tdTomato]Hze/J) mice were purchased from Jackson Laboratory (Bar Harbor, ME, USA). All mice were kept under pathogen-free conditions. To trace the lineage of *Sox9*-expressing cells in damaged tendons, we constructed double-transgenic *Sox9creERT2/Rosa26-loxP-stop-tdTomato* reporter (*Sox9-Cre ERT2*; *tdTomato*) mice in which Cre expression in Sox9-positive progenitor cells could be induced in different developmental stages by tamoxifen (Tam) administration.

### 4.2. Tendon Injury Mouse Model

For chronological evaluation following tendon injury, the mice were divided into four groups. As a control, sham surgery was performed in the Achilles tendon (sham), 1 week after the operation, 2 weeks after the operation, and 4 weeks after the operation (Figure 1A). We created a simple and highly reproducible partial excision model to minimize postoperative complications. A mixture of medetomidine (0.75 mg/kg), midazolam (4.0 mg/kg), and butorphanol (5.0 mg/kg) was administered to 6-week-old mice to induce deep anesthesia. The procedure adopted was the same as that described by Sakabe et al. (2018) [16]. Briefly, to expose the Achilles tendon, the anterior outer skin was incised approximately 1 cm slightly above the calcaneal bone. Subsequently, the Achilles tendon was partially excised (0.3 mm in width) using a no. 14 surgical blade (FEATHER Safety Razor Co., Ltd., Osaka, Japan) at approximately 2 mm proximally from the calcaneal insertion site while avoiding massive hemorrhage (Figure 1B). Then, the skin was closed using a 5-0 Ethicon suture (Ethicon^TM^, Raritan, NJ, USA). To promote swift arousal following the surgery, the mice were administered atipamezole (0.75 mg/kg) and returned to the cages. The mice were kept in standard cages until the samples were collected. One operator performed all surgical procedures to improve consistency. For cell lineage tracing, the mice were administered tamoxifen (dissolved in corn oil at 60 °C for 2 h; T5648-1G, Sigma-Aldrich, St. Louis, MO, USA) at a concentration of 20 mg/mL to induce Cre recombination. Tamoxifen was intraperitoneally injected at 100 L per injection using a 21-G syringe. By staggering the injection schedule, two groups were examined: Pre and Post. The Pre group received injections for 3 days starting 7 days before the Achilles tendon injury, while the Post group received injections for 4 days starting 1 day before surgery. In the uninduced control, only corn oil was injected [23].

### 4.3. Physiological Functional Measurements

For the chronological evaluation of functional recovery following tendon injury, the mice were examined following the same procedures described by Itoh (2017) [48]. Briefly, to induce deep anesthesia, a mixture of medetomidine (0.75 mg/kg), midazolam (4.0 mg/kg), and butorphanol (5.0 mg/kg) was administered to mice. Then, to attach a surface electrode (custom made; Bioresearch Center, Nagoya, Japan), the fur in the triceps surae muscle area was removed. To measure the maximum isometric torque induced by ankle dorsiflexion using a system consisting of an electric stimulator (SEN-3301; Nihon Kohden, Tokyo, Japan) and a pressure sensor, a viscous conductive gel (CR-S; Sekisui Plastics Co., Ltd., Osaka, Japan) was applied between the electrode and the skin. The electrode was fixed with adhesive tapes on the surface of the aponeurosis junction and the proximal 6 mm trajectory. Then, the isometric plantar flexion torque (T) was calculated from the pressure applied to the footplate (F) and the distance between the ankle axis and the sensor (r) as follows: T = Fr. Considering the fatigue caused by electrical stimulation, three measurements were made at 2 min intervals in the sham, 1 week after the operation, 2 weeks after the operation, and 4 weeks after the operation (Figure 1C).

### 4.4. Morphological Examination

All collected tissues were processed at room temperature. The tissue surrounding the Achilles tendon was immersed and fixed in 4% paraformaldehyde phosphate-buffered saline (PBS) solution. The samples were decalcified for 14 days using 10% ethylenediaminetetraacetic acid. Dehydration in 70% ethanol for 3 weeks was followed by replacement with paraffin over 1 day. The paraffin blocks were prepared as per the standard procedure, and using a sliding microtome (Leica Biosystems, Wetzlar, Germany), a series of 5 μm thick tissue segments were excised as frontal sections from the posterior calcaneal bone. The samples were stained with H&E and Azan and observed under an upright microscope (Axio Imager.2; Carl Zeiss Microscopy, Jena, Germany). The imaged tissue was analyzed using ImageJ Version 1.53t (NIH, Bethesda, MD, USA, http://rsbweb.nih.gov/ij/ accessed on 5 August 2022). The visualized tissue was analyzed using ImageJ. Briefly, the area of the tendon injury was measured. The area occupied by cells accumulated in the injured area was then measured and the ratio of the area occupied by cells in the injured area was assessed as the cell ratio.

### 4.5. RT-PCR

The part 1 mm above the calcaneal bone and the part 2 mm below the proximal muscle–tendon junction were resected to collect tendinous tissue. Only the injured site was selected using a microscope (Stemi305; Carl Zeiss Microscopy, Jena, Germany). The sham was not used to confirm the essential mRNA of the tendon. In particular, those without epithelial incisions were used as the control. The collected tissue was then used as the sample. The mRNA was extracted using RNAiso Plus (Takara Bio, Shiga, Japan). To collect cDNA at the injured site, cDNA was prepared using a QuantiTect Reverse Transcription Kit (Qiagen, Hilden, Germany). After optimum PCR conditions were determined for all primers, an experiment was conducted using a 7500 Fast Real-Time PCR System (Applied Biosystems, Foster City, CA, USA) to quantify RNA. The experiment was conducted according to the standard protocol for the 7500 Fast Real-Time PCR System. A Real-Time PCR Master Mix (Toyobo, Osaka, Japan) was used as a hot-start PCR reaction mix for the 7500 Real-Time PCR System. For each target gene, the PCR product contained 7.6 μL of sterilized water, 10.0 μL of qPCR, 0.4 μL of ROX reference dye, and 1.0 μL of TaqMan probe. In addition, 1.0 μL of diluted control cDNA was added to obtain a final reacting volume of 20.0 μL. The FAM-labeled TaqMan probe contained Sox9 (Mm00448840_m1), α-SMA (Mm00725412_s1), Postn (Mm00450111_m1), Collagen I (Mm00801666_g1), Collagen III (Mm00802305_g1), and fibromodulin (Mm00491215_m1). The gene expression levels were normalized against glyceraldehyde-3-phosphate dehydrogenase (GAPDH) as a control. Table 1 lists the primer assay IDs for real-time PCR and amplicon lengths.

### 4.6. Immunohistochemical Analysis

To analyze the protein expression in the sample, α-SMA markers related to the epitenon involved in tendon regeneration alongside Sox9 were evaluated [49]. The segments were immersed in hyaluronidase (25 mg/mL) and left for 45 min. After washing with PBS, they were incubated in 3% bovine serum albumin at 37 °C for 30 min to block nonspecific binding. Subsequently, the segments were incubated overnight at 4 °C in primary Sox9 antibody (1:1000 dilution; Merck Millipore, Burlington, MA, USA), α-SMA (1:200 dilution; ab5694, Abcam, Cambridge, UK), Postn (1:250 dilution; ab14041, Abcam), or an equivalent concentration of rabbit unconjugated anti-mouse polyclonal IgG as an isotype control (ab37415, Abcam). As a secondary antibody, the segments were dyed at 20 °C for 1 h using an ABC staining kit (Funakoshi, Tokyo, Japan). Some segments were treated with ImmPACT 3,3-diaminobenzidine (Funakoshi, Tokyo, Japan) to detect reactions. Before testing, Sox9 was contrast-stained with light green and α-SMA and Postn with hematoxylin. The segments were visualized using an upright microscope (Axio Imager.2; Carl Zeiss Microscopy, Germany). An analysis was performed for the visualized tissue using ImageJ. Briefly, the tendon injury site in proportion to the Sox9-, α-SMA-, and Postn-positive areas was evaluated as density.

### 4.7. Lineage Tracing Analysis

The sample was fixed in 4% PFA for 2 days, placed in a refrigerator, and then immersed in 10% sucrose, 20% sucrose, and 30% sucrose for 3 h each. Then, it was embedded in Super Cryoembedding Medium (CECTION-LAB: C-EM001) and instantaneously frozen in isopentane, which was cooled with liquid nitrogen. Moreover, 10 m thick segments were prepared using a cryostat (Leica Biosystems, Wetzlar, Germany). Kawamoto method tape (Cryofilm type 2C [9], 2.0 cm C-FP093) was used for sectioning [50]. Toluidine blue staining was performed to confirm the injury site. For tdTomato fluorescence evaluation, the sample was washed with 1X PBS and then enclosed in Fluoroshield Mounting Medium with DAPI (AR-6501-01).

### 4.8. Statistical Analysis

All statistical analyses were performed using R version 4.1.3. After confirming that the data were normally distributed, homoscedastic data were statistically analyzed using one-way analysis of variance as per Tukey’s multiple comparison test, and nonhomoscedastic data were statistically analyzed using the Games–Howell test. For all data, statistical significance was set at a difference of *p* < 0.05 (* *p* < 0.05, ** *p* < 0.01, and *** *p* < 0.001). Error bars indicate the standard deviation of the mean.

## 5. Conclusions

The results of this study revealed that Sox9 was expressed within tendons concurrently with the pre-structure of the epitenon, an important region during tendon regeneration and linked to functional regeneration. Moreover, the lineage tracing of Sox9 expressed during tendon regeneration revealed that cells that switched into Sox9-expressing cells following tendon injury were involved in regeneration. These results indicate that Sox9 plays an important role in tissue regeneration and functional restoration during tendon recovery. Our research team is conducting further investigations to ascertain the origin of Sox9 switching and accumulation in the injury site.

## Figures and Tables

**Figure 1 ijms-24-11305-f001:**
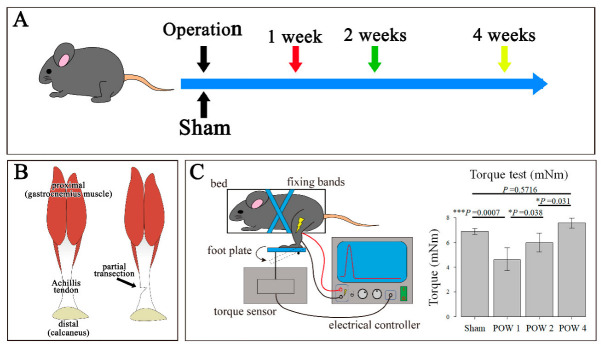
Tendon injury mouse model and functional torque tests. (**A**) Mice were subjected to tendon injury (*n* = 15) and sham surgery (sham) (*n* = 5), and tissue samples were collected 1 week after the operation (*n* = 5), 2 weeks after the operation (*n* = 5), and 4 weeks after the operation (*n* = 5). (**B**) To prepare the tendon injury mouse model, the Achilles tendon of each mouse was injured by approximately 0.3 mm using a scalpel. (**C**) The analysis was conducted using an Ito-style torque testing device to confirm the functional regeneration process. A significant functional decline was observed in the 1 week after the operation compared with that in the sham. No significant functional differences were noted between the sham and 4 weeks after the operation. * *p* < 0.05, *** *p* < 0.001.

**Figure 2 ijms-24-11305-f002:**
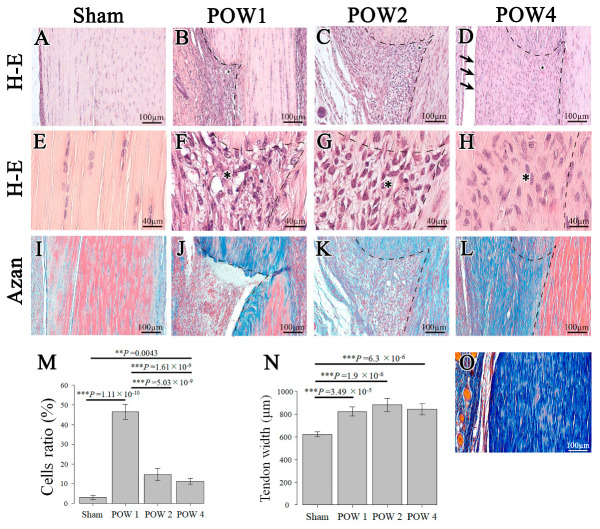
Morphological staining analysis (**A**–**D**). Hematoxylin and eosin (H&E) staining images at ×20 for morphological observation: (**A**) sham (*n* = 5), (**B**) 1 week after the operation (POW1; *n* = 5), (**C**) 2 weeks after the operation (POW2; *n* = 5), and (**D**) 4 weeks after the operation (POW4; *n* = 5). (**E**–**H**) H&E staining images at ×100 for cell polarity observation: (**E**) sham, (**F**) 1 week after operation (POW1), (**G**) 2 weeks after the operation (POW2), and (**H**) 4 weeks after the operation (POW4). (**I**–**L**) Azan staining was performed to investigate collagen fibers. (**M**) Proportion of cells in the injury site. (**N**) Width of the tendinous tissue. (**O**) Azan staining image of sham with excessive (24 h) staining with aniline blue. All segments were thinly sliced at 5 μm thickness, with the black dotted lines representing injury sites (**B**–**D**,**F**–**H**,**J**–**L**). Meanwhile, the black arrow indicates the pre-structure of the epitenon. Scale bars = 100 and 40 μm for the ×20 and ×100 magnified images. Asterisk indicates an enlarged section (*). The continuity of the epitenon that disappeared 1 week after the operation (**B**) was restored by 4 weeks after the operation (**D**). ** *p* < 0.01, *** *p* < 0.001.

**Figure 3 ijms-24-11305-f003:**
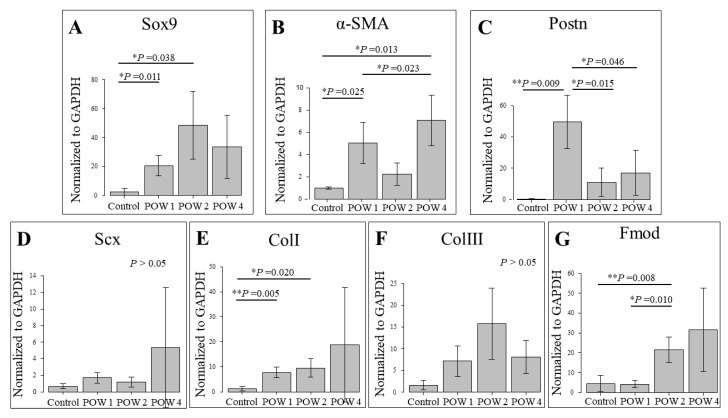
Real-time polymerase chain reaction. mRNA expression in the injury site was chronologically examined. The levels at the control (*n* = 5), 1 week after the operation (*n* = 5), 2 weeks after the operation (*n* = 5), and 4 weeks after the operation (*n* = 5) are listed. The analyzed mRNAs were (**A**) Sox9, (**B**) *α-*SMA, (**C**) Postn, (**D**) Scx, (**E**) collagen 1a1, (**F**) collagen 1a1, (**F**) collagen, and (**G**) fibromodulin. For all graphs, the significance level was set at * *p* < 0.05. In Sox9, significant differences were noted between the control and 1 week after the operation/2 weeks after the operation. Significant differences in Postn, α-SMA, and Col1 were noted. Fibromodulin, an essential substance in forming collagen fiber, increased significantly in the 2 weeks after the operation compared to the control. * *p* < 0.05, ** *p* < 0.01.

**Figure 4 ijms-24-11305-f004:**
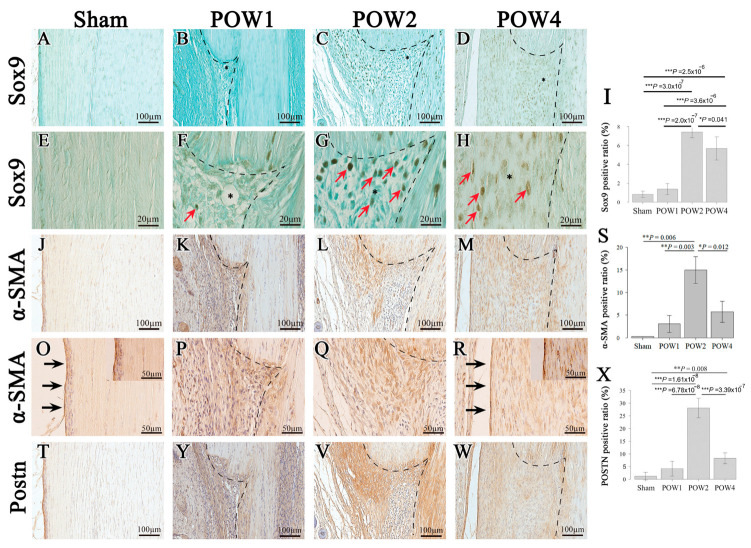
Immunohistochemical staining and analysis. To trace protein expression following Achilles tendon injury, immunohistochemical staining was performed. (**A**–**D**) Sox9 images at ×20 magnification. (**E**–**H**) Sox9 images at ×100 magnification. (**I**) Protein content in the injury site. (**J**–**M**) α-SMA at ×20 magnification. (**O**–**R**) α-SMA at ×40 magnification ((**O**,**R**) top right show ×100 images). (**S**) Protein content. (**T**,**V**,**W**) Postn at ×20 magnification. (**X**) Protein content. The vertical axis shows the sham (*n* = 5) (**A**,**E**,**J**,**O**,**T**), 1 week after the operation (*n* = 5) (**B**,**F**,**K**,**P**,**Y**), 2 weeks after the operation (*n* = 5) (**C**,**G**,**L**,**Q**,**V**), and 4 weeks after the operation (*n* = 5) (**D**,**H**,**M**,**R**,**W**) from top to bottom. All segments were thinly sliced with 5 μm thickness, with the black dotted lines indicating the injury sites. The red arrows in the ×100 magnified images indicate Sox9-postive cells (**F**–**H**, red arrows). Black arrows indicate epitenon or pre-structure of the epitenon stained positive for α-SMA ((**O**,**R**) black arrows). Scale bars = 100 and 50 and 20 μm for the ×20 and ×40 and ×100 magnified images, respectively. Asterisk indicates an enlarged section (*). For all graphs, the significance level was set at * *p* < 0.05. ** *p* < 0.01, *** *p* < 0.001.

**Figure 5 ijms-24-11305-f005:**
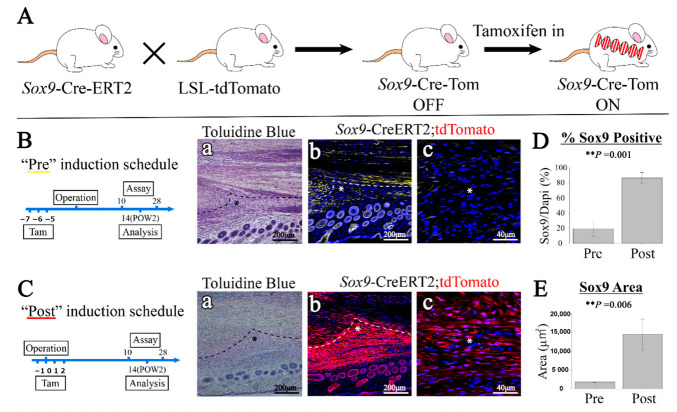
Lineage tracing of *Sox9*-expressing cells. Lineage tracing was performed for *Sox9*-expressing cells during tendon injury. (**A**) Schematic of hybridizing *Sox9-Cre-ERT2* mice and *LSL-tdTomato* mice to create *Sox9-Cre ERT2*; *tdTomato* mice (*n* = 6). (**B**) Schedule and staining images in the Pre group (*n* = 3). (a) Toluidine blue staining images. (b) Fluorescence staining images of *tdTomato*^+^ cells. (c) Magnified fluorescence staining images of *tdTomato*^+^ cells. (**C**) Schedule and staining images in the Post group (*n* = 3). (a) Toluidine blue staining images. (b) Fluorescence staining images of tdTomato^+^ cells. (c) Magnified fluorescence staining images of tdTomato^+^ cells. (**D**) Proportion of *Sox9-CreERT2*; *tdTomato*^+^ cells in all cells in the injury site. (**E**) Area of *Sox9-Cre ERT2*; *tdTomato*^+^ cells in the injury site. (**B**) The stained areas were colored yellow later to clarify the differences with (**C**). The dotted lines in (**B**,**C**) mark the boundaries of the injury site. In (a,b), scale bar = 200 μm, and in (c), scale bar = 40 μm. For all graphs, the significance level was set at ** *p* < 0.01.

**Table 1 ijms-24-11305-t001:** Primer assay ID and amplicon length of RT-PCR.

Primer		Assay ID	Amplicon Length
Sox9	transcription factor for differentiated chondrocyte and tendon cells	Mm00448840_m1	101
α-SMA	protein belonged actin, a marker of epitenon important during tendon regeneration	Mm00725412_s1	95
Postn	protein expressed in the extracellular matrix that supports TSPC growth and tendon formation	Mm00450111_m1	79
Scx	transcription factor for differentiated tendon cells	Mm01205675_m1	59
Col1a1	fibroplastic collagen abundant in tendons	Mm00801666_g1	89
Col3a1	early differentiation marker for regenerate injured tendons	Mm00802305_g1	80
Fmod	small leucine-rich proteoglycan for regulation in collagen matrix assembly	Mm00491215_m1	75
GAPDH	housekeeping genes expressed in most cells are used as an internal control	Mm99999915_g1	107

## Data Availability

RNA-sequencing data that support the findings of this study have been deposited in the DDBJ Sequenced Read Archive under the accession number DRA010453.

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
