# Peer review of "Chronological Changes in the Expression and Localization of Sox9 between Achilles Tendon Injury and Functional Recovery in Mice"

_ijms, 2023, doi:10.3390/ijms241411305_

Round 1

Reviewer 1 Report

The authors present an interesting study investigating the expression and localization of different proteins during the tendon healing process. The animal model is adequate and the results are interesting and nicely presented in the graphs and figures. My major concern, however, is the reliability of the data. They mention five animals per time point but performed different methods that required separate tissue processing. Based on the description it is not clear how many samples were used for histology and PCR and if the sample size allows a sound statistical analysis.

1.      Title: the title should be more precise. It should be mentioned that an animal model was used to investigate tendon healing.

2.      Abstract: the background is rather long, while the own study with the results is short. Please shorten the background and give more details on the animal model and the results.

3.      Introduction: the statement that no study investigated Sox9 during tendon regeneration is not correct. See 2023 DOI: 10.1302/2046-3758.125.BJR-2022-0340.R2, 2020 DOI: 10.1371/journal.pone.0242286, 2018: DOI: 10.1016/j.arthro.2017.10.045,. Please adjust your statement and discuss the studies.

4.      Results: please explain all abbreviations before first usage, e.g. POW

5.      Lines 91-94: this summary is not necessary. It just repeats the results from the lines before.

6.      Figure legends: please provide the sample size in all figure legends and for all methods

7.      I suggest to write e.g. “..one week after operation…”  instead of POW1 group. This will make the reading of the text easier. They investigated the effect of healing time and not the effect of different groups.

8.      Line 104, Fig 2C: can they really say that this structure is the epitenon? I suggest using a more carful formulation, such as “a prestructure of the epiteneon”? Same with POW4

9.      Fig. 2M: they show cell ratio %. What does this mean? Description line 410-11 is unclear

10.   Line 120, 136: Azan stains collagen but provides no information on collagen orientation.

11.   L 141-142: this sentence is unclear and which statistical test was made? Correlation analysis?

12.   L167-168: please check the sentence it is unclear. Increase at POW2 compared to POW1 and CTRL! Please check the description on the data in the figure legends.

13.   2.4: please rename the heading. They describe more than only Sox9

14.   L 170 ff: the description of the immunohistology seems a little bit confusing. What do they exactly mean with “injury site”? They see the formation of new tissue and tissue regeneration “within” the area of the cut tendon. I would call this injury site, therefore the one positive cells seems to be in the injury site (Fig 4F). Please clarify.

15.   aSMA is also a marker for myofibroblasts in tendons and ligaments (DOI: 10.1016/S0736-0266(01)00109-7).

16.   The description of the quantification of the IHC could be clearer and shorter: e.g. The amount of positive area increased from the intact tissue until week two after injury with a decreased at week four.

17.   Some formulations are unclear: e.g. Line 247 ff ….indicating that Sox9 expression infiltrated the injury site over time.” “Expression of Sox9”  cannot infiltrate tissue

18.   Linage tracing discussion: they see yellow stained cells in the injury site of the pre-group. The red staining in the post-group, however, is more intense. But does this really mean that all the positive cells were cells migrated into the injury site? It could also be that Sox9 expression is induced after injury in the cells that show a basal expression in the pre-group or in cells that were not expressing Sox9 in the pre-group.

19.   I am not convinced by the presented data that the epitenon recovered. This must be shown by a better analysis.

20.   L 283 ff: please provide a reference for the statement of high a low molecular weight discrimination by the staining.

21.   “cell polarity”: do they mean cell circularity or shape? Please explain when used for the first time.

22.   Animal model: they used 5 mice per group/time point. Due to the different methods used (histology, PCR), the tissue must have been split or the animal tissue used for different methods. This needs a clear description and might results in a sample size that allows no sound statistical analysis. I doubt that the data were normally distributed (Line 464), please show all data points in the graphs.

23.   Please add the data availability statement

see comments above

Author Response

Dear Reviewer

Thank you very much for reviewing our manuscript and offering valuable advice.

We have addressed your comments with point-by-point responses, and revised the manuscript accordingly.

Thank you very much for your excellent suggestion.

In the present experimental group, both legs underwent operation or Sham (pseudo-operation). Initially, functional tests were performed on the left leg in deep anesthesia.

Mice were then euthanized by CO2 intoxication and the left leg was harvested for PCR and the right leg for histological retrieval.

We have added the corrections in red in 4.1. Experimental Animals. Please check the details. L:368-372

  1. Title: the title should be more precise. It should be mentioned that an animal model was used to investigate tendon healing.

Thank you for your comment.

The title has been changed to “Chronological changes in expression and localisation of Sox9 from Achilles tendon injury to functional recovery in mice” to be more accurate. Thank you for your kind attention.

  1. Abstract: the background is rather long, while the own study with the results is short. Please shorten the background and give more details on the animal model and the results.

Thank you for your recommendation.

We have shortened the background and used red text to add additional information on the animal model and results. Please check the information.

 L:29-36

  1. Introduction: the statement that no study investigated Sox9 during tendon regeneration is not correct. See 2023 DOI: 1302/2046-3758.125.BJR-2022-0340.R2, 2020 DOI: 10.1371/journal.pone.0242286, 2018: DOI: 10.1016/j.arthro.2017.10.045,. Please adjust your statement and discuss the studies.

Thank you for your comment.

Thank you very much for providing us with this valuable paper. We have checked the paper and found that it is not strictly about tendon tissue, but about the regeneration process in the bone-tendon junction region and ligament region, which is called enthesis. In our search, we found few papers that focus on Sox9 in the regeneration process of tendon tissue alone.

We have revised the text and added references based on the references we received. Please check the information.

L:81-83

  1. Results: please explain all abbreviations before first usage, e.g. POW

Thank you for your kind suggestion. 

The official name is given before the abbreviation is used. Please make sure that you are aware of this.

L:91-94

  1. Lines 91-94: this summary is not necessary. It just repeats the results from the lines before.

Thank you for your comment.

Lines 91-94 have been deleted. Please check the information.

  1. Figure legends: please provide the sample size in all figure legends and for all methods

Thank you for your comment.

All Figure legends are marked with the number of samples in red.

Please check the information provided.

  1. I suggest to write e.g. “..one week after operation…”  instead of POW1 group. This will make the reading of the text easier. They investigated the effect of healing time and not the effect of different groups.

Thank you for your recommendation. The official name is given in red before the abbreviation is used. Instead of ”POW”, the notation has been changed to ”week after operation”.Please make sure that you are aware of this.

  1. Line 104, Fig 2C: can they really say that this structure is the epitenon? I suggest using a more carful formulation, such as “a prestructure of the epiteneon”? Same with POW4

Thank you for your recommendation

2.2. the text on epitenon in Morphological Analysis has been corrected in red to prestructure of the epitenon, as you indicated. We thank you for your kind attention.

L:111-167

  1. Fig. 2M: they show cell ratio %. What does this mean? Description line 410-11 is unclear

Thank you for your recommendation

The text in lines 410-11 has been corrected. Please check the text.

L:435-436

  1. Line 120, 136: Azan stains collagen but provides no information on collagen orientation.

Thank you for your comment.

The text in lines 120 and 136 has been corrected in red. Please check the information.

L:130-139

  1. L 141-142: this sentence is unclear and which statistical test was made? Correlation analysis?

Thank you for your comment.

We have removed the text you pointed out as it was unclear. Please take a moment to review it.

  1. L167-168: please check the sentence it is unclear. Increase at POW2 compared to POW1 and CTRL! Please check the description on the data in the figure legends.

Thank you for your comment.

The sentence you pointed out was unclear and has been corrected in red. Thank you for your comment.

L:186-188

  1. 2.4: please rename the heading. They describe more than only Sox9

Thank you for your comment.

We have changed the name of the heading. Please check the name of the new headline.

L:189

  1. L 170 ff: the description of the immunohistology seems a little bit confusing. What do they exactly mean with “injury site”? They see the formation of new tissue and tissue regeneration “within” the area of the cut tendon. I would call this injury site, therefore the one positive cells seems to be in the injury site (Fig 4F). Please clarify.

Thank you for your comment.

L 170ff: Although the expression was not as pronounced within the tendon injury site, the strongest expression was observed outside the tendon injury site (Figure 4B and 4F; red arrows).The sentence 'Sox9-positive cells were found to be expressed at the injury site in POW1' may have caused confusion. However, we stated this because more Sox9-positive cells were expressed in areas of connective tissue than in injury sites. We will delete this information as it may cause confusion to the reader. We thank you in advance for your confirmation.

L:192-193

  1. aSMA is also a marker for myofibroblasts in tendons and ligaments (DOI: 10.1016/S0736-0266(01)00109-7).

Thank you for your valuable paper.

We have added the information you indicated in L 272-273. We have also added the references.

L284-286

  1. The description of the quantification of the IHC could be clearer and shorter: e.g. The amount of positive area increased from the intact tissue until week two after injury with a decreased at week four.

Thank you for your recommendation

We have made it shorter and clearer as you indicated. Please take a moment to check it.

L:224-228

  1. Some formulations are unclear: e.g. Line 247 ff ….indicating that Sox9 expression infiltrated the injury site over time.” “Expression of Sox9”  cannot infiltrate tissue

Thank you for your comment.

You pointed out Line 247 ff ....indicating that Sox9 expression infiltrated the injury site over time."

has been corrected to read "increased" rather than "infiltrated". Thank you for your kind attention.

L:261-264

  1. Linage tracing discussion: they see yellow stained cells in the injury site of the pre-group. The red staining in the post-group, however, is more intense. But does this really mean that all the positive cells were cells migrated into the injury site? It could also be that Sox9 expression is induced after injury in the cells that show a basal expression in the pre-group or in cells that were not expressing Sox9 in the pre-group.

Thank you for your comment.

As you point out, our discussion states that Sox9 expression is induced after injury in the cells that show a basal expression in the pre-group or in cells that were not expressing Sox9 in the pre-group.

We discuss this in paragraph 8 of the Discussion. We thank you for your kind attention.

L:350

  1. I am not convinced by the presented data that the epitenon recovered. This must be shown by a better analysis.

Thank you for your comment.

The evaluation of epitenon was carried out using α-SMA based on previous reports of specific accumulation of α-SMA in the epitenon region. Results of immunostaining data are presented in 2.4. Immnohistochamical staining in Tendon Injury section. Please check the information in this section.

  1. L 283 ff: please provide a reference for the statement of high a low molecular weight discrimination by the staining.

As described in the paper, a commercially available staining solution is used, so I didn't understand "the statement of high a low molecular weight discrimination by the staining". very sorry.

  1. “cell polarity”: do they mean cell circularity or shape? Please explain when used for the first time.

Thank you for your comment.

As you pointed out, we have used "cell polarity" for the first time in 2.2. Morphological Analysis and have provided an explanation. Please check it.

L:123-126

  1. Animal model: they used 5 mice per group/time point. Due to the different methods used (histology, PCR), the tissue must have been split or the animal tissue used for different methods. This needs a clear description and might results in a sample size that allows no sound statistical analysis. I doubt that the data were normally distributed (Line 464), please show all data points in the graphs.

Thank you for your comment.

We have made a more detailed review of the areas you pointed out in 4.1, Experimental Animals. Please check the details in 4.1.

L:368-372

  1. Please add the data availability statement

Thank you for your comment.

Data availability

RNA-sequencing data that support the findings of this study have been deposited in the

DDBJ Sequenced Read Archive under the accession number DRA010453.

The following information has been added. Please check the following information.

Reviewer 2 Report

The authors studied the involvement of SOX9 in tendon regeneration using mouse model.

Comments

1.      Section 2.1 or 4.2: The authors should describe POW groups and their numbers.

2.      On each Figure (from 1 to 5) the number of animals should be indicated.

3.      Results section: The authors should not repeat in the text the p-values presented on graphs. This should be corrected.

4.      Discussion should not contain references on Figures. This should be corrected.

5.      Lines 116; 118-119; 320;379; 410-411; 450-451; 456-457: These sentences are not clear. They should be clarified.

6.      Line 335: The authors should indicate what kind of fracture they mean.

7.      Line 360: Reference or detailed protocol should be presented.

Lines 116; 118-119; 320;379; 410-411; 450-451; 456-457: These sentences are not clear. They should be clarified.

Author Response

Dear Reviewer

Thank you very much for reviewing our manuscript and offering valuable advice.

We have addressed your comments with point-by-point responses, and revised the manuscript accordingly.

  1. Section 2.1 or 4.2: The authors should describe POW groups and their numbers.

Thank you very much for your excellent suggestion.

Five 6-week-old male C57BL/6J mice (Sankyo Labo Service Corporation, Inc., Tokyo, Japan) were assigned to each group. In the present experimental group, both legs were operated or Sham surgery was performed. Initially, functional tests were performed on the left leg of the mice in deep anesthesia. Mice were then euthanized with CO2 poisoning and the left leg was harvested for PCR and the right leg was harvested for histological analysis.

We have added the corrections in red in 4.1. Experimental Animals. Please check the details.

L:368-372

  1. On each Figure (from 1 to 5) the number of animals should be indicated.

Thank you for your comment.

All Figure legends are marked with the number of samples in red.

Please check the information provided. Figure 1-5

  1. Results section: The authors should not repeat in the text the p-values presented on graphs. This should be corrected.

Thank you for your comment.

All p-values mentioned in the text have been deleted.

Please check the information in the text.

  1. Discussion should not contain references on Figures. This should be corrected.

Thank you for your comment.

We have corrected the references you indicated. Please check them carefully.

  1. Lines 116; 118-119; 320;379; 410-411; 450-451; 456-457: These sentences are not clear. They should be clarified.

Thank you for your comment.

We have corrected the references you indicated. Please check them carefully.

L116;L123-125

L 118-119; L126-129

L320;L339-342

L379;L404-407

L410-411;L435-436

L450-451;L479-480

L456-457;L486-487

  1. Line 335: The authors should indicate what kind of fracture they mean.

Thank you for your comment.

We have described the fracture you mentioned more specifically in which part of the body the fracture is located. We thank you for your kind attention.

L354-356

  1. Line 360: Reference or detailed protocol should be presented.

 Thank you for your comment.

We have attached the references for the protocols you mentioned. Please take a moment to review it.

L391-397

Comments on the Quality of English Language

Lines 116; 118-119; 320;379; 410-411; 450-451; 456-457: These sentences are not clear. They should be clarified.

Thank you for your comment.

We have corrected the quality of the English text once again. Please check the quality of the English text.

Round 2

Reviewer 1 Report

The authors addressed my comments. Unfortunately, I am not satisfied with the answers and the changes made. The authors made several textual changes. However, there are a lot of mistakes and incorrectness in the new text passages. I suggest that an English scientific writer reads and corrects the manuscript. 

Comment 7: they considered my comment (partially) and changed the group description. I suggest to delete “group” after the time points because it is not necessary for the understanding but makes reading more difficult.

e.g. Line 91ff: The physiological test revealed a significant decline in functions one week after operation (POW1) compared with those in the sham group. Two weeks after operation (POW2), the mice exhibited a significant increase in function compared to mice one week after operation. Please check and modify this in the entire document. 

Comment 8: „ ..prestructure of epitenon…“ . I suggest that they use this formulation in the entire document, meaning also in the abstract, discussion and conclusion when they describe this structure.

Comment 9: I am sorry but this is still unclear. Figure mentions “cells ratio %”, but the added description in M&M is “Briefly, tendon injury sites were selected and the percentage of cells accumulated was assessed as density.” How can a cell number be expressed in % by this analysis? This needs a better description.  

Comment 19 Epitenon: In Fig 2c,d they indicate with arrows the “new epitenon”. Looking at Fig 4l,m, these structures are not a-sma positive. Therefore, I am still not convinced that they see a recovery of the epitenon. It is unclear, which structure they mean with “new epitenon”. 

Comment 20: The Azan stain does not discriminate the molecular weight of the stained structures. It used dyes of different molecular weight. This must be clarified!!! Furthermore, the tendon should stain blue and not red. The authors must explain, why the intact tendon stains primarily red! Please check the description of the staining and your stained tissues. 

 The authors made several textual changes. However, there are a lot of mistakes and incorrectness in the new text passages. I suggest that an English scientific writer reads and corrects the manuscript.

Author Response

Dear Reviewer 1,

We thank you again for reviewing our manuscript and for the valuable advice provided. We have made corrections regarding the points raised by you. Please also note that an English scientific writer has read the manuscript and made corrections.

Comment 7: they considered my comment (partially) and changed the group description. I suggest to delete “group” after the time points because it is not necessary for the understanding but makes reading more difficult.

e.g. Line 91ff: The physiological test revealed a significant decline in functions one week after operation (POW1) compared with those in the sham group. Two weeks after operation (POW2), the mice exhibited a significant increase in function compared to mice one week after operation. Please check and modify this in the entire document. 

We thank the reviewer for this comment. As suggested, we have deleted all mentions of “group” after the time point.

Comment 8: „ ..prestructure of epitenon…“ . I suggest that they use this formulation in the entire document, meaning also in the abstract, discussion and conclusion when they describe this structure.

We thank the reviewer for this helpful recommendation. All text has been corrected in red to read “pre-structure of the epitenon.” Please check the information carefully.

Comment 9: I am sorry but this is still unclear. Figure mentions “cells ratio %”, but the added description in M&M is “Briefly, tendon injury sites were selected and the percentage of cells accumulated was assessed as density.” How can a cell number be expressed in % by this analysis? This needs a better description.  

We thank the reviewer for this comment. We apologize for the inappropriate description. In this analysis, the cell count was not expressed as a %. In fact, the area of the tendon injury was measured. The area occupied by the cells that accumulated in the injured area was then measured, and the ratio of the area occupied by the cells in the injured area was assessed as a cell ratio. Please note that we have introduced the corrections in red.

L: 436–438

Comment 19 Epitenon: In Fig 2c,d they indicate with arrows the “new epitenon”. Looking at Fig 4l,m, these structures are not a-sma positive. Therefore, I am still not convinced that they see a recovery of the epitenon. It is unclear, which structure they mean with “new epitenon”. 

We thank the reviewer for the attention paid to this matter. The staining images included in Figures 2 and 4 are serial sections of the same individual cut at 5-µm intervals for each week, to ensure linkage. At 2 weeks after operation, the black arrow in Figure 2C was deleted because, as stated by the reviewer, the outer part of the tendon was not a-sma positive. The description of the pre-structure of the epitenon at 2 weeks after operation has also been deleted. We believe that the lateral part of the tendon was a-sma positive at 4 weeks after operation; therefore, we have added a 40x image of a-sma staining in Figure 4. We believe that the outer part of the tendon is the “new epitenon.” We have also added a reference in the Materials and Methods pertaining to the specific expression of a-sma in the epitenon.

L:109-111 L:469-470

Comment 20: The Azan stain does not discriminate the molecular weight of the stained structures. It used dyes of different molecular weight. This must be clarified!!! Furthermore, the tendon should stain blue and not red. The authors must explain, why the intact tendon stains primarily red! Please check the description of the staining and your stained tissues. 

We thank the reviewer for this comment. According to Lee et al, the native tendon stains red. We also agree with that result. When glass was dipped in aniline blue for 10 min, the native tendon stained red. We consider that the color depends on the time of the dip in to the stain. We added new photos of a long-term dipping glass in Figure 2O. The description of the annotation of the molecular weight in the Azan staining experiment has also been corrected. Please check the details.

L: 296–302

Comments on the Quality of English Language

 The authors made several textual changes. However, there are a lot of mistakes and incorrectness in the new text passages. I suggest that an English scientific writer reads and corrects the manuscript.

We have made corrections regarding the points raised by you. Please note that an English scientific writer has read the manuscript and made corrections.